# Metabolic Phenotypes—The Game Changer in Quality of Life of Obese Patients?

**DOI:** 10.3390/healthcare10040617

**Published:** 2022-03-25

**Authors:** Ivona Mitu, Cristina Preda, Cristina Daniela Dimitriu, Ovidiu Mitu, Irina Iuliana Costache, Manuela Ciocoiu

**Affiliations:** 1Department of Morpho-Functional Sciences II, University of Medicine and Pharmacy “Grigore T. Popa”, 700115 Iasi, Romania; ivona.mitu@umfiasi.ro (I.M.); daniela.dimitriu@umfiasi.ro (C.D.D.); manuela.ciocoiu@umfiasi.ro (M.C.); 2Department of Endocrinology, University of Medicine and Pharmacy “Grigore T. Popa”, 700115 Iasi, Romania; cristina.preda@umfiasi.ro; 31st Medical Department, University of Medicine and Pharmacy “Grigore T. Popa”, 700115 Iasi, Romania; irina.costache@umfiasi.ro

**Keywords:** obesity phenotypes, quality of life, fat mass, metabolic syndrome

## Abstract

Background: The present study aimed to investigate the association of obesity phenotypes and quality of life (QoL) scales and their relationship with fat mass (FM) parameters. Methods: This study categorized 104 subjects into 4 obesity phenotypes based on BMI and metabolic syndrome status: metabolically healthy obese (MHO), metabolically unhealthy obese (MUO), metabolically healthy non-obese (MHNO), and metabolically unhealthy non-obese (MUNO). Body composition was measured by dual-energy X-ray absorptiometry (DEXA) and metabolic profile was characterized by blood samples. All subjects completed the SF-36 item Short Form Health Survey Questionnaire. Results: Comparing the four obesity phenotypes, significant results were reported for Bodily Pain between MHNO/MUNO (*p* = 0.034), for Vitality between MHO/MUO (*p* = 0.024), and for Mental Component Score between MHO/MUO (*p* = 0.026) and MUO/MUNO (*p* = 0.003). A more thorough inside-groups analysis yielded a positive and moderate to high correlation between FM parameters and QoL scales in MHO and MHNO, while a negative and weak to moderate correlation was observed in MUO and MUNO. Conclusion: This study reported an inverse U-shaped relationship between FM and QoL in obesity phenotypes, suggesting that metabolic status is a key factor involved in modulating QoL and therefore challenging the idea of obesity as a main driver of low QoL. We recommend the inclusion of FM percentage in the definition of obesity phenotypes in future research, to better evaluate QoL of obesity phenotypes.

## 1. Introduction

Health-related quality of life (HRQoL) is a concept used to define a patient’s subjective perspective on his own physical, mental, and social functioning [1]. It is important to understand disease impact on HRQoL as this will help in the decision making of physicians, policymakers, and even patients [1,2]. Obesity represents one of the major health concerns worldwide and needs to be treated as a complex disease with numerous mechanisms and etiologies; its consequences need to be understood, including its impact on quality of life [3].

A growing body of research continues to demonstrate associations between obesity and reduced quality of life [2,4], but analysis performed on specific obesity phenotypes is rare and reports controversial results. A past meta-analysis showed that physical HRQoL was impaired among overweight and obese subjects, whereas mental HRQoL was reduced only among subjects with type III obesity (BMI ≥ 40 kg/m^2^) [4]. At the same time, a study on 5608 participants, conducted by the same authors, showed that both metabolically unhealthy and healthy individuals have low QoL, suggesting that “healthy obesity” is a misnomer [5].

The classification of patients based on obesity and metabolic status reveals specific obesity phenotypes, from which the “metabolically healthy obese” (MHO) phenotype is the most attractive one, promising a healthier status of the patient disregarding the high BMI, when compared to metabolically unhealthy obese (MUO) (i.e., lower risk of cardiometabolic events [6,7], type 2 diabetes [8], hypertension [8]). A recent prospective study of 381,363 UK Biobank participants reported that MHO have higher incident rates than non-obese subjects for heart failure and respiratory disease, but not for fatal atherosclerotic cardiovascular disease. The same study identified a lower risk of all-cause mortality for MHO than MUO [9]. These solid findings support further research between phenotypes within the obese, overweight, or normal-weight group, separately.

In this study we aimed to characterize obesity phenotypes and explore the association (especially of MHO and MUO) with quality of life scales and scores. In addition, investigating whether obesity (defined by BMI) or metabolic status weights more in lowering quality of life could provide more evidence on this important topic. Another milestone for our study was to establish whether BMI or fat mass parameters were better associated with a change in quality of life of our subjects, considering the complex relationship between BMI and HRQoL [10,11].

## 2. Materials and Methods

### 2.1. Study Design and Participants

This cross-sectional cohort study included 104 subjects that respected the inclusion and exclusion criteria, narrowed down from 474 consecutive patients visiting the Cardiology Clinic from a university hospital in Iasi, Romania, for the first time or for follow-up. Eligible participants were in the 35–75 age group and had no known chronic disease or had not followed treatment in the last 6 months for any cardiovascular or metabolic disease, no antecedent atherosclerotic acute pathology, and no current pregnancy in women. The study was conducted over a period of 2 years (between 2020–2022) and approved by the University Ethics Committee, number 1 on 27 July 2020. All subjects have agreed and signed an informed consent in order to take part in this study.

### 2.2. Definition and Measurements

#### 2.2.1. Clinical and Anthropometric Measurements

Patients had their weight, height, abdominal circumference, hip circumference, pulse, oxygen saturation, abdominal and tricipital skinfold thickness (ST) measured. All parameters except the ones resulting from dual-energy X-ray absorptiometry (DEXA) were measured two times, by the same person, during the entire study. Participants were wearing light clothing and they had not engaged in physical exercise at least 30 min prior to investigations. Weight and all fat mass parameters were measured using DEXA. Height was measured using a tape meter stadiometer while subjects were standing on bare feet with head, shoulders, buttocks, and heels leaning against a surface that was at a 90° angle to the floor. ST was measured using a Holtain-type caliper, following the correct way to approach the skinfold: for tricipital ST, halfway between the acromion process and olecranon process, and for abdominal ST, 5 cm lateral of the umbilicus [12]. A flexible tape was used to measure abdominal circumference at the umbilical level and hip circumference at the greater trochanter level [13].

Resting heart rate and oxygen saturation were measured with a pulse oximeter. Blood pressure was registered twice with a validated automatic device and cuffs of three sizes, according to arm circumference, after a 15 min rest in the seated position.

#### 2.2.2. Biochemical Measurements

Blood samples were collected by trained personnel after a 12 h overnight fast and analyzed in the same day. The measurements of glucose, total cholesterol (CHOL-T), high-density lipoproteins (HDL), and triglycerides (TG) were performed in the same laboratory, each time using the same technique, for all patients. HDL by the direct method (elimination/catalase), TG by the glycerol phosphate oxidase method, and glucose by the glucose-6-phosphate dehydrogenase method.

#### 2.2.3. Obesity Phenotypes

Participants in the study were categorized into four different phenotypes: metabolically healthy obese (MHO), metabolically unhealthy obese (MUO), metabolically healthy non-obese (MHNO), and metabolically unhealthy non-obese (MUNO). Several definitions exist for metabolic health and the most used one in studies [14,15,16] reports the presence of abdominal circumference > 88 cm (women)/104 cm (men) and maximum 1 abnormal component from the metabolic syndrome criteria: glucose ≥ 100 mg/dL, HDL < 40 mg/dL (men)/< 50 mg/dL (women), TG ≥ 150 mg/dL, SBP/DBP ≥ 130/85 mmHg [17]. Metabolically unhealthy status was established if patients presented a higher than normal abdominal circumference and the metabolic syndrome (≥2 abnormal components). Obesity was defined by a BMI ≥ 30 kg/m^2^.

### 2.3. Health Related Quality of Life (QoL)

The short-form 36 item questionnaire was used to assess quality of life among participants. This questionnaire has been previously studied in the Romanian population and its validity and reliability were approved [18]. The SF-36 form is a 36-item self-completed questionnaire exploring health across eight areas: physical functioning (PF), role limitation due to physical problems (RP), role limitation due to emotional problems (RE), social functioning (SF), mental health (MH), energy/vitality (V), bodily pain (BP), and general health (GH) perceptions. The results for the eight scales were obtained with the standardized principal component analysis method employing orthogonal rotation [19].

There are two more scores that evaluate globally the physical and mental health of the patient which were calculated with specific algorithms that involve the factor analysis of normative datasets gained from general population surveys. Normally, each population has its own factor weights to which the algorithm refers to calculate the physical component score (PCS) and mental component score (MCS), but for easier cross-cultural interpretation of data, studies should refer to US factor weights [20,21]. To calculate the values for PCS and MCS we standardized each of the eight scales by calculating a Z-score (subtracting scale mean from the Romanian population norm for that scale and dividing the result by the standard deviation of the Romanian mean for that scale [18]). Afterward, Z-scores were multiplied by factor weights from the US population (a more accurate future comparison with other studies implies reporting to the same standard US values as mentioned in other papers) for PCS, respectively MCS [20,22], and summed over all eight scales. Finally, these scores were standardized to a T-score multiplying the results for PCS and MCS by 10 and adding 50 to the product (the mean was set to 50 and standard deviation to 10) [19,22,23].

### 2.4. Covariates

Covariates concerned in self-reported basic demographic data, cigarette smoking, alcohol consumption, risk of diabetes, obesity in childhood/family, and physical activity level. Smoking status was divided into three groups (current smoker, former smoker, and never smoked) and chronic alcohol consumption was positively assessed if subjects consumed more than 14 g/day for women and 28 g/day for men [24].

Physical activity was evaluated with the International Physical Activity Questionnaire (IPAQ). Patients reported how much time they spent walking or performing moderate and/or vigorous activities in the past week. The activity level was considered low, moderate, or high depending on the MET (metabolic equivalent of task) levels obtained from the 2000 compendium of physical activities [25,26].

The FINDRISK tool is widely used for predicting the risk of developing diabetes mellitus and includes the evaluation of age, BMI, waist circumference, physical activity, consumption of fruits/vegetables, blood pressure medication, previous high glucose blood levels, and family history of diabetes. The total risk score is considered very low risk if the score is <7 (an estimated 1 in 100 people will develop the disease), low risk if the score is between 7–11 (an estimated 1 in 25 people will develop the disease), moderate risk if the score is between 12–14 (an estimated 1 in 6 people will develop the disease), high risk if the score is between 15–20 (an estimated 1 in 3 people will develop the disease), and very high risk if the score is >20 (an estimated 1 in 2 people will develop the disease).

### 2.5. Statistical Analysis

No data substitution algorithms were adopted to impute missing data. Data analysis was conducted using SPSS version 23.0 (IBM Corporation, Armonk, NY, USA) and Microsoft Excel 2003 (Microsoft Corporation, Redmond, WA, USA). To assess the distribution of the variables we analyzed histograms, skewness, and kurtosis values [27] and performed Kolmogorov–Smirnov/Shapiro–Wilk tests. For normally distributed data we applied one-way ANOVA tests, and for non-normally distributed data we applied the Kruskall–Wallis test. The assumption of homogeneity of variances was tested with Levene’s F test and statistical power was expressed as eta squared or Cohen’s d value [28]. Results were considered statistically significant if *p* < 0.05.

Continuous variables were reported as means ± standard deviations or means and confidence intervals. Categorical variables were expressed as frequencies (percentages). The continuous and categorical variables between obesity phenotypes were compared with *t*-test/one-way ANOVA or Kruskall–Wallis and chi-square test, respectively. Correlations between continuous variables were described by Pearson’s r.

Analysis was conducted for the whole sample and repeated in phenotype-stratified subgroups.

## 3. Results

### 3.1. Baseline Characteristics of the Study Population

The distribution of sociodemographic factors, smoking status, chronic alcohol consumption, level of physical activity, risk of developing diabetes mellitus, and other factors is presented in Table 1. The MHO group represents 19.23% of the entire cohort and 30.77% of the obese participants. The prevalence of other obesity phenotypes is 43.27% for MUO, 15.38% for MHNO, and 22.12% for MUNO. Women were predominant in our study (74.04%). Smoking status was not significantly different between MHO and MUO even though we observed a higher percentage amongst the MUO. On the other hand, chronic alcohol consumption was significantly different between the two phenotypes (*p* = 0.022), with a higher degree of consumption for MUO (22.2% vs. 0%). Risk of diabetes was higher for MUO vs. MHO (*p* = 0.037) and for MUNO vs. MHNO (*p* = 0.027). The prevalence of a high level of physical activity is almost three times higher for MHO than for MUO.

The clinical and biochemical characteristics of the total study population according to metabolic health and obesity are reported in Table 2. Age was not significantly different between the subgroups of obese and non-obese. Fat mass percentage, abdominal and tricipital skinfold are higher in MHO than in MUO, while BMI mean is a bit higher in MUO than in MHO, suggesting more adipose tissue is present in MHO.

When comparing MHO with MHNO groups, none of the metabolic syndrome parameters were statistically significant, but when comparing MUO and MUNO groups there was a statistically significant result for HDL-cholesterol (*p* = 0.047, η^2^ = 0.06). Even though both groups are metabolically unhealthy, 6% variability in HDL-cholesterol is accounted for by the patient’s obese or non-obese phenotype.

### 3.2. SF-36 Scales and Scores in Obesity Phenotypes

The descriptive statistics associated with quality of life across the four obesity phenotypes are reported in Table 3. Prior to conducting any statistical test, the assumption of normality was evaluated and determined to be satisfied for GH and V scale and for PCS and MCS scores. Even though the distributions for all the eight scales and the two component scores were associated with skew and kurtosis less than |2.0| and |7.0|, respectively, the results for the test of normality (Kolmogorov–Smirnov, Shapiro–Wilk) confirmed a normal distribution only for GH, V, PCS, and MCS (Appendix A).

In order to test the hypothesis that inclusion in one particular phenotype is associated with a change in the quality of life of subjects, a one-way ANOVA test was performed for GH, V, MCS, PCS and Kruskall–Wallis test for all the other variables. All the statistically significant results are illustrated in Figure 1.

The assumption of homogeneity of variances for MCS score for the MHO and MUO groups was tested and satisfied based on Levene’s F test, F(1,63) = 0.499, *p* = 0.483. The independent between groups ANOVA for MCS score yielded a statistically significant effect, F(1,63) = 5.179, *p* = 0.026, η^2^ = 0.08, suggesting that MUO have a better mental quality of life than MHO. Thus, 8% of the variance in MCS score was accounted for by the MHO/MUO phenotype, with a moderate effect size based on Cohen’s guidelines (d = −0.59). The same analysis was performed for the MUO and MUNO groups F(1,66) = 9.461, *p* = 0.003, η^2^ = 0.12), concluding that 12% of the variance in MCS score was accounted for by the MUO/MUNO phenotype, with a moderate to high effect size based on Cohen’s guidelines (d = 0.74). Furthermore, MUO have a higher vitality score than MHO F(1,63) = 5.388, *p* = 0.024, η^2^ = 0.08, d = 0.60), with 8% of the variance in V scale being accounted for by MUO/MHO phenotype.

Results also report a 8.4% variability in rank scores for BP, which is accounted for by phenotype (*p* = 0.034, η^2^ = 0.084). A more thorough analysis reported a 18% variability in rank scores for BP accounted for by the inclusion of the subject in the MHNO or MUNO phenotypes (*p* = 0.008, η^2^ = 0.18) and 19% variability in rank scores for BP accounted for by the inclusion of the subject in the MHO or MHNO phenotypes (*p* = 0.01, η^2^ = 0,19).

### 3.3. Comparison between BMI and Fat Mass in Terms of QoL in Obesity Phenotypes

Fat mass parameters report a better association than BMI with the SF-36 questionnaire scales for the obesity phenotypes, but also for the entire study population. Regardless of the metabolic abnormalities, BMI significantly correlates only with PCS (r = −0.280, *p* = 0.004), whereas Trunk fat (kg) significantly correlates both with PCS (r = −0.301, *p* = 0.002) and with MCS (r = 0.221, *p* = 0.024). The results are under debate in the literature as to whether both scores go down in obese people by definition or it depends on the comorbidities associated. Therefore, we further analyzed each phenotype concerning the parameters for quality of life (Table 4).

Of all the subjects in the study, the ones included in the MHO phenotype present a better and more consistent correlation between QoL scales and fat mass/percentage. The correlation is always positive and moderate, suggesting that for obese subjects with no metabolic syndrome, the more adipose tissue they accumulate, the better the quality of life is, both physical (PF, BP) and mental (MH, MCS).

The MUO present weak and negative correlations between PF/PCS score and obesity parameters, implying that subjects with a higher fat mass have a lower quality of life from a physical point of view.

MHNO subjects report a positive and moderate to high correlation between fat mass and SF. They present a better quality of life, but only from a social point of view.

A moderate and negative significant correlation is observed in the MUNO group, where PF, RP and RE show a lower quality of life in patients with higher mass of adipose tissue.

## 4. Discussion

This is the first study to our knowledge that compares the power of BMI to the power of fat mass parameters to assess quality of life in obesity phenotypes. Interestingly, all our subjects were considered abnormal from the fat mass perspective, having higher than normal values for fat mass percentage (>25% for men, >32% for women), even though 37.5% of participants were non-obese accordingly to BMI (26.99 ± 2.24). Our analysis reported that parameters for adipose mass evaluate better than BMI the quality of life of obese subjects and of obesity phenotypes. Fat mass percentage, trunk mass percentage, and tricipital skinfold report correlations between all four obesity phenotypes and all scores of the SF-36 questionnaire, except GH and V. Regarding correlations between BMI and SF-36 scores, only the MUO group reported a significant negative correlation between BMI and PCS. These results support the findings of a 10,133 participants study that describes a nuanced relationship between BMI and HRQoL, challenging the idea of obesity defined by BMI as a main driver of reduced HRQoL [10]. Moreover, considering that we reported statistically different values only for MCS and BP between all four groups, fat mass percentage is to be accounted for when analyzing QoL in obesity phenotypes, as it is the only parameter that correlates with both.

Our findings showed that metabolically unhealthy subjects have a lower quality of life once they accumulate more adipose tissue, but when considering metabolically healthy phenotypes, obese and non-obese, a significant better quality of life is associated with more fat mass. Practically, we observed an inverted U-shaped relationship between fat mass and quality of life. As they gain more fat mass, metabolically healthy subjects have a better quality of life, that will decrease once they become unhealthy. This data suggests that metabolic status is a relevant factor involved in the modulation of HRQoL of patients.

Obesity phenotypes are not clearly defined in the literature [29,30,31]. There are studies that use homeostatic model assessment for insulin resistance (HOMA-IR) and BMI to classify the phenotypes [31], but most studies use the metabolic syndrome and BMI to define them [14,15,16]. The definition for metabolic syndrome proposed by the National Cholesterol Education Program-Third Adult Treatment Panel (NCEP-ATP III) was also used in our study [17].

The MHO phenotype remains a controversial scientific issue, because even though MHO are more healthy than other phenotypes it is not clear if later in life they preserve the healthy status or they become unhealthy [32]. Our study found one MHO person in every 3.25 individuals with obesity installed (BMI ≥ 30 kg/m^2^). The prevalence of MHO in studies varies from 6–75% and depends on the criteria implemented for MHO definition [33]. Philips et. al analyzed various studies and reported similar results concerning the prevalence of MHO in obese individuals when considering the same definition for MHO-30.2% vs. 30.7% in our study [34,35].

Similar to our finding, other studies also reported non-normal distribution for almost all sf-36 scales [36,37], and one study even concluded that the choice of statistical approach had no influence on the results, recommending statistical parametric tests no matter the distribution of data [38]. Since our analysis did not involve adjusting for covariates, we decided to use both parametric and non-parametric tests.

Our results suggest that obese patients with metabolic syndrome have a better mental QoL than MHO and MHNO. The effect size for this analysis is moderate to high and the variance in MCS score is rather significant (8% and 12%). Vitality also has higher scores in MUO than MHO, supporting the better mental QoL of MUO. Most studies report a lower physical score in obesity in general, but contrary to our findings they report no statistical significance for the mental score [39,40,41]. On the other hand, Donini LM et al found no statistical significance for both physical and mental scores when comparing MHO and MUO [42] and Lopez-Garcia et al. found similar MCS scores among all obesity phenotypes [43]. Our scores for PCS are approximately the same for MHO vs. MUO and lower than the ones for the non-obese phenotypes, but not statistically significant. We can conclude that in our study only the mental component is statistically different between obesity phenotypes.

As expected, in the non-obese groups MHNO subjects present better scores on BP than MUNO (*p* < 0.05), but when comparing obese to non-obese, the mean value for BP in MUO is greater than the one in MUNO (*p* > 0.05). This last result, though not statistically significant, suggests a better general perception of pain in the unhealthy obese group and is in accordance with the results reported by Price et al. This study shows that obese subjects are less sensitive to pain in the area with excessive subcutaneous fat, such as the abdominal area [44]. This may be the result of decreased nerve fiber density and increased anti-inflammatory cytokines in adipose tissue [44,45].

The strength of the study is its thorough analysis of each HRQoL scale and score amongst the obesity phenotypes to identify in which specific groups does the variance of a particular scale occur and to determine the degree of correlations inside each phenotype. Besides the BMI, our study provides information about fat mass parameters as a tool for better characterizing body mass. Furthermore, all anthropometric parameters were measured by standard protocols and trained staff and the SF-36 questionnaire used to assess HRQoL is a tool already validated in this population. A few limitations have to be acknowledged. The small sample group did not provide the possibility to further classify the non-obese group into metabolically healthy/unhealthy overweight/normal-weight groups. A recent review reported the importance of early detection of metabolic obesity in patients with normal weight, in order to avoid undesirable consequences such as atherosclerosis, coronary artery disease or diabetes. Therefore, investigating this phenotype may represent another focus of our future research [46]. Moreover, no comparisons based on gender or age groups were performed and adjusting for confounders was difficult since the study cohort was relatively small.

## 5. Conclusions

This study reported an inverse U-shaped relationship between fat mass and quality of life concerning the obesity phenotypes, suggesting that metabolic status is a key factor involved in modulating HRQoL and therefore challenging the idea of obesity as a main driver of low HRQoL. Furthermore, we recommend the inclusion of fat mass percentage in the definition of obesity phenotypes in future research, to better evaluate quality of life of obesity phenotypes.

## Figures and Tables

**Figure 1 healthcare-10-00617-f001:**
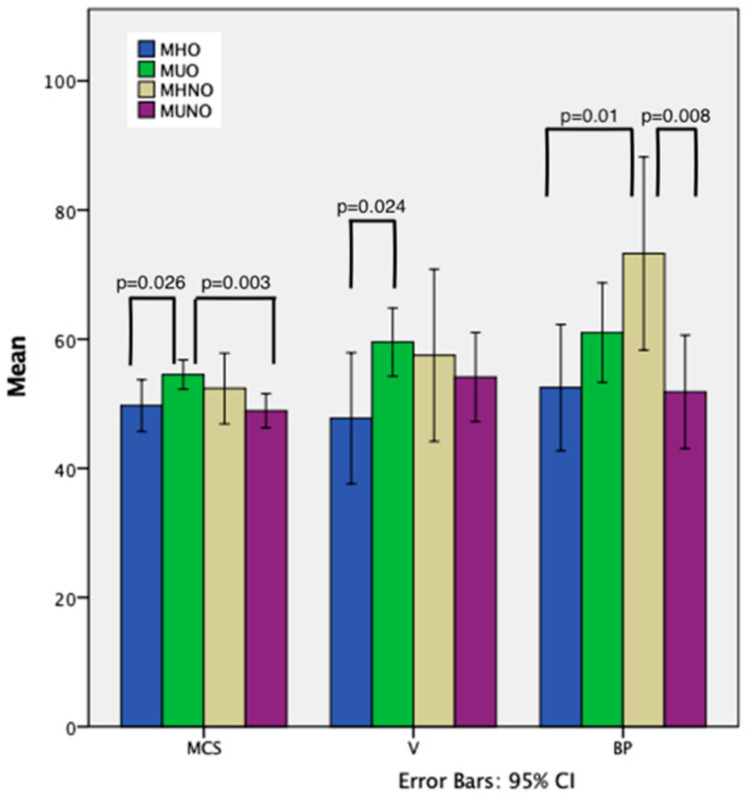
Mental Component Score (MCS), Vitality (V) and Bodily Pain (BP) means values across obesity phenotypes.

**Table 1 healthcare-10-00617-t001:** Baseline characteristics of obesity phenotypes.

	Obese Patients			Non-Obese Patients		
	MHO (n = 20)	MUO (n = 45)	*p*	MHNO (n = 16)	MUNO (n = 23)	*p*
*Gender*			0.047			0.627
Females	85% (17)	60% (27) *		81.2% (13)	87% (20)	
Males	15% (3)	40% (18) *		18.8% (3)	13% (3)	
*Residence*			0.757			0.987
Urban	65% (13)	68.9% (31)		56.3% (9)	65.3% (13)	
Rural	35% (7)	31.1% (14)		43.8% (7)	36.5% (10)	
*HCA*			0.083			0.960
Yes	25% (5)	8.9% (4)		12.5% (2)	13% (3)	
No	75% (15)	91.1% (41)		87.5% (14)	87% (20)	
*Menopause* *(only women)*			0.523			0.751
Yes	82.4% (14)	74.1% (20)		92.3% (12)	95% (19)	
No	17.6% (3)	25.9% (7)		7.7% (1)	5% (1)	
*Smoking status*			0.219			0.217
Current	10% (2)	17.8% (8)		12.5% (2)	4.3% (1)	
Former	20% (4)	35.6% (16) *		6.3% (1) *	26.1% (6)	
Never	70% (14)	46.7% (21) *		81.3% (13)	69.6% (16)	
*Chronic alcohol consumption*			* **0.022** *			0.492
Yes	0% (0) *	22.2% (10) *		6.3% (1)	13% (3)	
No	100% (20) *	77.8% (35) *		93.8% (15)	87% (20)	
*FINDRISK score*			* **0.037** *			* **0.027** *
Very low	0% (0)	0% (0)		18.8% (3) *	4.3% (1)	
Low	40% (8)	11.1% (5) *		75% (12) *	39.1% (9)	
Moderate	30% (6)	31.1% (14) *		0% (0) *	13% (3)	
High	30% (6)	51.1% (23) *		6.3% (1) *	39.1% (9) *	
Very high	0% (0)	6.7% (3)		0% (0)	4.3% (1)	
*IPAQ score*			* **0.027** *			0.571
Low	35% (7)	31.1% (14) *		50% (8)	47.8% (11) *	
Moderate	20% (4) *	51.1% (23) *		43.8% (7)	34.8% (8)	
High	45% (9) *	17.8% (8)		6.3% (1)	17.4% (4)	
*Childhood obesity*			0.632			0.282
Yes	15% (3)	20% (9)		37.5% (6)	21.7% (5)	
No	85% (17)	80% (36)		62.5% (10)	78.3% (18)	
*1st degree relatives with obesity*			0.865			0.501
Yes	60% (12)	62.2% (28)		50% (8)	39.1% (9)	
No	40% (8)	37.8% (17)		50% (8)	60.9% (14)	

* adjusted residual for the respective value is larger than 1.96, indicating that the number of cases is significantly (*p* < 0.05) different than would be expected if the null hypothesis were true. MHO = metabolically healthy obese, MUO = metabolically unhealthy obese, MHNO = metabolically healthy non-obese, MUNO = metabolically unhealthy non-obese, HCA = heredocolateral antecedents (cardiovascular events in first degree relatives), FINDRISK = Finnish Diabetes Risk Score, IPAQ = International Physical Activity Questionnaire.

**Table 2 healthcare-10-00617-t002:** Clinical and biochemical characteristics of obesity phenotypes.

	Obese Patients			Non-Obese Patients		
	MHO	MUO	*p*/η^2^	MHNO	MUNO	*p*/η^2^
Age	56.65 ± 7.36	58.4 ± 8.58	0.432	61 ± 7.51	63.13 ± 7.48	0.388
BMI	34 ± 3.68	34.96 ± 4.89	0.437	26.11 ± 2.36	27.61 ± 1.98	***0.039*** η^2^ = 0.11
AC	108.22 ± 11.62	115.08 ± 11.98	***0.036*** η^2^ = 0.07	95.56 ± 6.49	101.65 ± 6.42	***0.006*** η^2^ = 0.18
HC	117.25 ± 7.33	120.02 ± 10.49	0.289	104.81 ± 6.24	106.83 ± 4.62	0.254
Abdominal skinfold	39.9 ± 7.12	37.69 ± 7.51	0.270	30.13 ± 7.71	34.39 ± 8.24	0.111
Tricipital skinfold	29.05 ± 6.56	27.91 ± 7.86	0.574	24.38 ± 7.80	25.52 ± 6.60	0.624
Total Fat (kg)	38.78 ± 7.23	40.02 ± 11.82	0.664	26.82 ± 7	29.83 ± 4.70	0.117
Total Lean and BMC (kg)	52.92 ± 10.96	59.03 ± 10.84	***0.040*** η^2^ = 0.06	44.19 ± 5.38	45.60 ± 8.10	0.548
Fat mass (%)	42.41 ± 4.21	40.21 ± 6.82	0.189	37.47 ± 7.70	39.73 ± 5.48	0.291
Trunk fat mass (kg)	19.21 ± 5.48	21.28 ± 7.52	0.272	12.44 ± 3.67	14.90 ± 2.86	***0.024*** η^2^ = 0.13
Trunk fat mass (%)	41.67 ± 4.86	41.23 ± 6.57	0.789	36.46 ± 7.95	39.96 ± 5.33	0.107
SBP	129.05 ± 18.81	143.73 ± 17.92	***0.004*** η^2^ = 0.12	135.12 ± 28.09	135.87 ± 16.47	0.918
DBP	80.95 ± 10.94	89.11 ± 10.12	***0.005*** η^2^ = 0.12	84.69 ± 15.33	84.96 ± 12.78	0.953
Glucose	94.45 ± 11.29	120.67 ± 34.92	***0.002*** η^2^ = 0.15	99.69 ± 19.33	121.09 ± 50.27	0.115
TG	97.65 ± 26.46	195.71 ± 95.81	***<0.001*** η^2^ = 0.24	99.69 ± 30.42	163.13 ± 54.60	***<0.001*** η^2^ = 0.32
HDL-Chol	56.9 ± 11.31	46.87 ± 13.34	***0.005*** η^2^ = 0.12	62.25 ± 12.94	53.52 ± 11.76	***0.035*** η^2^ = 0.11

MHO = metabolically healthy obese, MUO = metabolically unhealthy obese, MHNO = metabolically healthy non-obese, MUNO = metabolically unhealthy non-obese, BMI = body mass index, AC = abdominal circumference, HC = hip circumference, BMC = bone mineral content, SBP = systolic blood pressure, DBS = diastolic blood pressure, TG = triglycerides. Reference interval: SBP ≤ 130 mg/dL, DBP ≤ 85 mg/dL, Glucose ≤ 100 mg/dL, TG ≤ 150 mg/dL, HDL-Chol < 40 mg/dL (men)/HDL-Chol < 50 mg/dL (women).

**Table 3 healthcare-10-00617-t003:** Quality of life scales and scores across obesity phenotypes.

	Obese Patients					Non-Obese Patients					*p* for All Groups Comparison
	MHO		MUO			MHNO		MUNO			
	Mean (95% CI)	SD	Mean (95% CI)	SD	*p*	Mean (95% CI)	SD	Mean (95% CI)	SD	*p*	
PF	58.25 (45.79–70.71)	26.62	55.78 (48.32–63.24)	24.84	0.648 ^a^	63.75 (49.27–78.23)	27.17	59.57 (46.33–72.8)	30.6	0.606 ^a^	0.715 ^a^
RP	41.25 (20.71–61.79)	43.89	47.78 (36.14–59.42)	38.74	0.417 ^a^	62.5 (40.21–84.79)	41.83	52.17 (33.19–71.16)	43.89	0.421 ^a^	0.452 ^a^
BP	52.5 (42.71–62.29)	20.92	61 (53.3–68.7)	25.62	0.272 ^a^	73.28 (58.32–88.24)	28.07	51.85 (43.08–60.62)	20.28	** *0.008* ** ^a^	***0.034*** ^a^
GH	49.5 (40.08–58.92)	20.12	52.67 (47.7–57.63)	16.53	0.508 ^b^	59.69 (47.33–72.05)	23.2	51.3 (42.24–60.37)	20.95	0.247 ^b^	0.436 ^b^
V	47.75 (37.61–57.89)	21.67	59.56 (54.27–64.85)	17.6	** *0.024* ** ^b^	57.5 (44.16–70.84)	25.03	54.13 (47.24–61.02)	15.93	0.610 ^b^	0.150 ^b^
SF	68.13 (57.65–78.6)	22.39	78.06 (72.8–83.32)	17.5	0.078 ^a^	82.81 (71.97–93.66)	20.35	71.2 (62.5–79.89)	20.10	0.078 ^a^	0.098 ^a^
RE	48.33 (26–70.67)	47.73	61.48 (49.41–73.55)	40.17	0.272 ^a^	68.75 (43.24–94.26)	47.87	47.82 (28.96–66.68)	43.61	0.162 ^a^	0.320 ^a^
MH	64.8 (54.52–75.08)	21.97	70.4 (65.71–75.09)	15.60	0.388 ^a^	63.75 (48.85–78.65)	27.96	59.13 (52.08–66.18)	16.13	0.366 ^a^	0.100 ^a^
PCS	43.16 (39.32–47)	8.20	43.44 (41.14–45.75)	7.68	0.893 ^b^	48.42 (43.65–53.19)	8.95	45.14 (41.05–49.22)	9.44	0.283 ^b^	0.191 ^b^
MCS	49.71 (45.7–53.72)	8.57	54.51 (52.25–56.77)	7.52	** *0.026* ** ^b^	52.37 (46.89–57.85)	10.28	48.93 (46.27–51.58)	6.13	0.198 ^b^	**0.026** ^b^

^a^ Kruskall–Wallis test, ^b^ One-way Anova test, MHO = metabolically healthy obese, MUO = metabolically unhealthy obese, MHNO = metabolically healthy non-obese, MUNO = metabolically unhealthy non-obese, PF = physical function, RP = role limitation due to physical problems, BP = bodily pain, GH = general health, V = vitality, SF = social functioning, RE = role limitation due to emotional problems, MH = mental health, PCS = physical component score, MCS = mental component score.

**Table 4 healthcare-10-00617-t004:** Correlations between quality of life scales and weight/obesity parameters.

		BMI	Total Fat (kg)	Fat%	Trunk Fat (kg)	Trunk%	Abdominal Skinfold	Tricipital Skinfold
PF	r p	NS	^4^ −0.462 0.027	^1^ 0.438 0.05	NS	NS	NS	^2^ −0.298 0.047
RP	r p	NS	NS	NS	NS	^4^ −0.450 0.031	NS	NS
BP	r p	NS	NS	^1^ 0.476 0.034	NS	NS	NS	NS
GH	r p	NS	NS	NS	NS	NS	NS	NS
V	r p	NS	NS	NS	NS	NS	NS	NS
SF	r p	NS	^3^ 0.494 0.05	^3^ 0.508 0.044	^3^ 0.547 0.028	^3^ 0.591 0.016	NS	NS
RE	r p	NS	^4^ −0.420 0.046	^4^ −0.462 0.026	NS	^4^ −0.438 0.037	NS	NS
MH	r p	NS	NS	^1^ 0.479 0.033	NS	NS	NS	^1^ 0.438 0.05
PCS	r p	^2^ −0.308 0.04	^2^ −0.288 0.05	NS	^2^ −0.327 0.028	NS	NS	^2^ −0.307 0.04
MCS	r p	NS	NS	^1^ 0.474 0.035	NS	NS	^1^ 0.477 0.034	^1^ 0.434 0.05

^1^ = MHO (metabolically healthy obese), ^2^ = MUO (metabolically unhealthy obese), ^3^ = MHNO (metabolically healthy non-obese), ^4^ = MUNO (metabolically unhealthy non-obese), BMI = body mass index, PF = physical function, RP = role limitation due to physical problems, BP = bodily pain, GH = general health, V = vitality, SF = social functioning, RE = role limitation due to emotional problems, MH = mental health, PCS = physical component score, MCS = mental component score, NS = not significant, r = Pearson’s coefficient.

## Data Availability

Data supporting reported results are available from the corresponding authors.

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
