# Peer review of "Metabolic Phenotypes—The Game Changer in Quality of Life of Obese Patients?"

_healthcare, 2022, doi:10.3390/healthcare10040617_

Round 1

Reviewer 1 Report

The paper “Metabolic phenotypes - the game changer in quality of life of obese patients?” is interesting and well written, based on a sound methodology and very clearly presented.

However, I have some comments that I think can enhance the readability of the paper:

  • Line 68: it could be useful to clearly where is the “Cardiology Clinic” located. Is it in Romania?
  • Lines 128-129: the reason why standardised scores from Romania were multiplied by weights obtained from the US population could be better explained.
  • Table 1: it seems to me that the meaning of “AHC” is not explained anywhere.
  • Table 3: I think that the readability of the paper would increase indicating full names instead of acronyms (that are explained 4 pages before), or at least it is worth to put a note to the table with their meaning like in Table 2.

Author Response

Dear Reviewer,

In name of all authors, I kindly thank you for taking the time and reviewing our paper. We strongly appreciate the comments in the reviewer’s report and we conducted changes in the manuscript in order to address all the issues mentioned.

  • The hospital is a university hospital located in Iasi, Romania. We have added this information in line 68.
  • For easier cross-cultural interpretation of data our study included the US factor weights for calculating the values for Physical Component Summary and Mental Component Summary. Standard means and standard deviations were used from the Romanian population, but in order to provide results that can be compared with results from other studies we need a standard reference to report to. Therefore, this standard reference is considered in our study factor weights from the US population, where a great majority of studies are published. This idea is supported by the scientific literature (please see reference 20 and 21). We have included an explication regarding this in lines 131-133.
  • We have added abbreviations to all tables in order to simplify the readability and understanding of the data.

Once again thank you for your time and relevant remarks.

Reviewer 2 Report

It has been a privilege to be able to evaluate this interesting and well-designed work.

We need abbreviations to be put in the tables, especially in tables 1, 2 and 3, MHO, MUO, MHNO, MUNO, IPAQ, BMI are missing. And in table 3 and 4 all the abbreviations are missing. And for the figure 1.

The references should include Pluta, W.; DudziÅ„ska, W.; Lubkowska, A. Metabolic Obesity in People with Normal Body Weight (MONW)—Review of Diagnostic Criteria. Int. J. Environ. Res. Public Health 2022, 19, 624. https://doi.org/10.3390/ijerph19020624

Author Response

Dear Reviewer,

In name of all authors, I kindly thank you for taking the time and reviewing our paper. We strongly appreciate the comments in the reviewer’s report and we conducted changes in the manuscript in order to address all the issues mentioned.

  • We have added abbreviations to all tables in order to simplify the readability and understanding of the data.
  • The reference indicated is now included in the manuscript. Thank you for the recommendation.

Once again thank you for your time and relevant remarks.

Reviewer 3 Report

Dear Authors,

I have read the manuscript “Metabolic phenotypes - the game changer in quality of life of obese patients?” by I. Mitu et al. with a high attention. The authors have carefully collected data over two years, keeping many things in mind. It is not easy to find subjects who meet all the required conditions for two years. Therefore, it is understandable that the number has been reduced from 474 to 104 subjects. They also paid great attention to sampling to minimize the possibility of error or differences in sampling. Phenotyping parameters were clearly defined for both sexes. Self-assessment of quality of life related to respondent health was also seriously approached with a 36-item questionnaire that was approved by the experts in 2001. Attention is also given to cross-cultural interpretation of the data (taking into account weight factors from the USA). In addition to medical parameters, genetic predispositions and lifestyle habits of the subjects (smoking, alcohol, childhood obesity, ...) were also considered. All collected data were compared and statically processed. Based on the obtained data, the subjects were classified into four metabolic phenotype types according to their metabolic status and obesity status.

Now, I must emphasize that the manuscript was viewed from the perspective of chemist. Thus, the text was quite difficult to understand because of a large number of abbreviations and numerical values. I preferred to edit the data by age group, because someone who is in the group MHO in his thirties is very likely not to be in it in his fifties. For example, in paragraphs 299,300 it says, "Our study found one MHO person in 3.25 individuals with obesity." Are all 104 respondents included here?

However, numerous studies in the COVID-19 pandemic have shown that overweight and obese people have more severe disease, and since the fat mass parameter is more sensitive, it is important that the results presented here be published and that medical and nutrition professionals consider them in defining the phenotype of obesity and in treating their patients.

I recommend accepting the received paper in the Healthcare as such.

Author Response

Dear Reviewer,

In name of all authors, I kindly thank you for taking the time and reviewing our paper. We strongly appreciate the comments in the reviewer’s report and we conducted changes in the manuscript in order to address all the issues mentioned.

  • We have added abbreviations to all tables in order to simplify the readability and understanding of the data.
  • Unfortunately, our sample size did not allow for a more thorough analysis based on age or gender. I agree that a younger person is not as healthy as an older one, but I must emphasize that from 104 participants in the study, only 17 people were between 39-49 years old. From these 17 subjects, 4 people were Metabolically Healthy Obese.
  • "Our study found one MHO person in 3.25 individuals with obesity." refers only to subjects that are obese (BMI ≥ 30 kg/m2). We have added a remark in line 330 to better clarify this aspect.

Once again thank you for your time and relevant remarks.